# Chikungunya Virus RNA Secondary Structures Impact Defective Viral Genome Production

**DOI:** 10.3390/microorganisms12091794

**Published:** 2024-08-29

**Authors:** Laura I. Levi, Emily A. Madden, Jeremy Boussier, Diana Erazo, Wes Sanders, Thomas Vallet, Veronika Bernhauerova, Nathaniel J. Moorman, Mark T. Heise, Marco Vignuzzi

**Affiliations:** 1Viral Populations and Pathogenesis Unit, Department of Virology, Institut Pasteur, CNRS UMR 3569, 75015 Paris, Francemarco_vignuzzi@idlabs.a-star.edu.sg (M.V.); 2Infectious Disease Department, Université Paris Cité and Hôpital Saint-Louis and Lariboisière, APHP, INSERM U944, 75010 Paris, France; 3Department of Microbiology and Immunology, University of North Carolina at Chapel Hill, Chapel Hill, NC 27514, USA; 4A*STAR Infectious Diseases Labs (A*STAR ID Labs), Agency for Science, Technology and Research (A*STAR), 8A Biomedical Grove, Immunos #05-13, Singapore 138648, Singapore; 5Infectious Diseases Translational Research Programme, Yong Loo Lin School of Medicine, National University of Singapore, Singapore 117597, Singapore; 6Lineberger Comprehensive Cancer Center, University of North Carolina at Chapel Hill, Chapel Hill, NC 27599, USA; 7Department of Genetics, University of North Carolina at Chapel Hill, Chapel Hill, NC 27599, USA

**Keywords:** chikungunya virus, defective viral genome, RNA secondary structure, SHAPE-MaP

## Abstract

Chikungunya virus (CHIKV) is a mosquito-borne RNA virus that poses an emerging threat to humans. In a manner similar to other RNA viruses, CHIKV encodes an error-prone RNA polymerase which, in addition to producing full-length genomes, gives rise to truncated, non-functional genomes, which have been coined defective viral genomes (DVGs). DVGs have been intensively studied in the context of therapy, as they can inhibit viral replication and dissemination in their hosts. In this work, we interrogate the influence of viral RNA secondary structures on the production of CHIKV DVGs. We experimentally map RNA secondary structures of the CHIKV genome using selective 2′-hydroxyl acylation analyzed by primer extension and mutational profiling (SHAPE-MaP), which couples chemical labelling with next-generation sequencing. We correlate the inferred secondary structure with preferred deletion sites of CHIKV DVGs. We document an increased probability of DVG generation with truncations at unpaired nucleotides within the secondary structure. We then generated a CHIKV mutant bearing synonymous changes at the nucleotide level to disrupt the existing RNA secondary structure (CHIKV-D2S). We show that CHIKV-D2S presents altered DVG generation compared to wild-type virus, correlating with the change in RNA secondary structure obtained by SHAPE-MaP. Our work thus demonstrates that RNA secondary structure impacts CHIKV DVG production during replication.

## 1. Introduction

Chikungunya virus (CHIKV), which belongs to the alphavirus genus and Togaviridae family, is a positive-strand RNA virus. During its acute phase, CHIKV is responsible for a dengue-like syndrome associating brutal fever with symptoms such as severe joint pain or rash [1,2,3,4]. CHIKV infection can lead to years-long polyarthralgia which incapacitates patients and strongly impacts their quality of life [2,4,5,6]. CHIKV has been responsible for two worldwide epidemics since the beginning of the 21st century, affecting 60 countries and causing close to 8 million cases altogether [4,7,8]. As with most RNA viruses, the error-prone replication of CHIKV in infected cells leads to the production of defective viral genomes (DVGs) [9,10], which represent mutated, truncated or rearranged genomes. DVGs are unable to complete a full viral cycle but have been documented as influencing viral replication and the activation of the immune system [11,12,13]. Notably, they are strong inducers of pro-inflammatory cytokines, including type-I interferons, as documented during syncytial respiratory virus and influenza virus infections in animal models and patients [14,15,16,17]. Work on arboviruses confirmed that this effect on innate immunity also exists in insects, in which DVGs from Sindbis, chikungunya and Zika viruses can modulate antiviral immunity [9] and block viral dissemination and transmission in the mosquito vector [10,18]. Truncated DVGs are hypothesized to arise via non-homologous recombination occurring during viral replication [12,19], a mechanism by which the viral error prone RNA-dependent RNA-polymerase (RdRp) detaches from its genome template at a specific position (hereinafter referred to as “start breakpoint”) and reattach to another position further along in the genome (“stop breakpoint”). The DVG arising from such an event will be truncated for the portion of genome between the start and stop breakpoints. While the existence of DVGs has been documented in CHIKV infection [10], the factors influencing their production are currently unknown.

A key parameter that influences recombination by viral polymerases is the existence of secondary and tertiary structures in viral RNA, which are generated by RNA folding upon itself [19]. These structures are essential for the viral life-cycle, as they are involved in replication and packaging [20,21] via the presence of local secondary RNA structure (such as hairpins or stem loops) and long-range interactions [20,22]. For example, four stem-loops in the 5′-UTR and the start of NSP1 of the CHIKV genome are key for positive- and negative-strand RNA synthesis [23]. Several studies link RNA secondary structure to homologous copy-choice recombination events, notably, in human immunodeficiency virus (HIV) [24,25,26], brome mosaic virus [27], the poliovirus Sabin strain [28] and hepatitis delta virus [29]. By this mechanism, when replicating the viral genome, the RdRp drops off the RNA template it started copying and reattaches to a second RNA template, giving rise to a hybrid RNA molecule. Because it detaches and reattaches at the same position on the RNA, the recombination is called homologous, and gives rise to a hybrid full-length RNA that is still infectious. Overall, the existence of secondary RNA structures is thought to shape the homologous recombination hotspots in the viral genome. Yet only a few studies have examined how the link between secondary RNA structures and non-homologous recombination might affect DVG formation. In at least one example, namely, Cymbidium ringspot virus, a highly base-paired region of a long DVG was thought to direct generation of a shorter DVG [30].

RNA secondary-structure analyses traditionally relied on thermodynamics-based, computer-aided structural predictions to determine the structure with the minimum free energy for folding, corresponding to the highest stability. These methods, however, must still be confirmed by experimental data. The technique of selective 2′-hydroxyl acylation analyzed by primer extension and mutational profiling (SHAPE-MaP) is an improvement, one which allows the creation of experimentally informed RNA secondary-structure models [31,32,33]. SHAPE-MaP relies on selectively acetylating unpaired nucleotides, i.e., bases that are not involved in RNA secondary structures. The acetylated nucleotides are then identified as mutations by next-generation sequencing [34]. From the mutation data, each nucleotide position is assigned a SHAPE reactivity value: a high value if the nucleotide is paired, and low, if the nucleotide is more likely to be unpaired. SHAPE-MaP can thus construct an experimentally-driven quantitative map of RNA secondary structure which depicts the probability that a given nucleotide is involved in RNA secondary structures. SHAPE-MaP was used to decipher the full secondary structure at a single-nucleotide level for several alphavirus RNA genomes, including Sindbis virus (SINV) [23], Venezuelan equine encephalitis virus (VEEV) [23] and, more recently, CHIKV [35].

In this work, we exploit the SHAPE-MaP of the CHIKV genome to interrogate the influence of RNA secondary structures on non-homologous recombination events leading to DVG generation. We show that, in infected mammalian cells, the higher the probability that a given nucleotide is unpaired, the higher the probability of it being a DVG breakpoint. To experimentally verify this correlation, we generate a CHIKV mutant, termed CHIKV D2S for “disrupted secondary structures”, which carries 76 synonymous mutations that abolish mapped RNA secondary structures in the first half of the CHIKV genome. We observe that, although CHIKV D2S generates DVGs from the same genomic regions as wild-type (WT) CHIKV, DVGs arising from D2S replication are more diverse in terms of sequences and display a decreased accuracy of breakpoint position. The differences between CHIKV WT and D2S were more important in the region of the genome that was disrupted in the secondary structure, compared to the undisrupted region. Importantly, we could correlate the nature of the DVGs produced by the D2S mutant to its RNA genomic structure determined by SHAPE-MaP, directly implicating RNA secondary structures in the regulation of non-homologous recombination and DVG generation.

## 2. Materials and Methods

### 2.1. Cells and Virus

Vero and BHK cells were maintained in Dulbecco’s modified Eagle’s medium (DMEM) and supplemented with 10% fetal calf serum (FCS; Gibco, Thermo Fischer Scientific, Life Technologies Corporation, Grand Island, NY, USA), 1% non-essential amino-acid (NEAA; Gibco, Thermo Fischer Scientific, Life Technologies Corporation, Grand Island, NY, USA) and 1% penicillin/streptomycin (P/S Thermo Fisher Scientific, Life Technologies Corporation, Grand Island, NY, USA) in a humidified atmosphere, at 37 °C, with 5% CO_2_.

The viral stocks were generated from chikungunya virus (CHIKV) infectious clones derived from the Caribbean strain, Asian genotype (described in [36]) or the disrupted secondary structure (D2S) mutant derived from it (see below). Plasmids were linearized with Not I enzyme (Thermo Fisher) and in vitro transcripted with the SP6 mMESSAGE mMACHINE kit (Invitrogen, Thermo Fischer Scientific, Life Technologies Corporation, Grand Island, NY, USA). RNA from in vitro transcription (IVT) was then transfected in BHK cells using lipofectamin 2000 (Invitrogen, Thermo Fischer Scientific, Life Technologies Corporation, Grand Island, NY, USA) and passaged once in Vero cells. The stocks were titered and kept at −80 °C before use. For D2S, the stocks were RNA-extracted and Sanger-sequenced to confirm the absence of reversion.

### 2.2. Cloning Selected Disrupted Secondary Structure (D2S) Mutant

Three regions of the original Caribbean CHIKV clone were mutated using the program CodonShuffle with the dn231 algorithm [37] in order to generate maximum secondary-structure disruption with minimum change in codon usage.

Five 154- to 214-nucleotide-long double-stranded DNA (gBlocks^®^ Gene Fragments, Integrated DNA Technologies, Coralville, Iowa, USA) carrying the wanted mutations (Appendix A) were ordered from IDT. Gene fragments were amplified by PCR using Q5 DNA polymerase (NEB). They were then cloned into WT CHIKV Carib plasmid one after the other, amplifying the CHIKV Carib vector around the region of modification using Q5 DNA polymerase (NEB); a list of primers used is available in Appendix A. After DpnI (Thermo Fisher Scientific, Life Technologies Corporation, Grand Island, NY, USA) treatment and gel purification (Macherey Nagel PCR and gel purification kit, Düren, Nordrhein-Westfalen, Germany) of the vector, insert and vector were fused together with In-Fusion reagent (Takara Bio Reagent) following the manufacturer’s instructions; 2.5 μL was transformed in XL10-Gold extra competent cells (Stratagene, La Jolla, CA, USA). Colonies were grown in LB medium with ampicillin, minipreps were performed (NucleoSpin plasmid, Macherey Nagel, Düren, Nordrhein-Westfalen, Germany) and Sanger-sequenced to confirm that the colonies carried the expected mutations.

### 2.3. Plaque Assay

Viral titration was performed on confluent Vero cells plated in 24-well plates one day before infection. Ten-fold dilutions were performed in DMEM alone and transferred onto Vero cells. After allowing infection for one hour, DMEM with 2% FCS, 1% P/S, 1% NEAA and 0.8% agarose was added on top of the cells. Three days post-infection, Vero cells were fixed with 4% formalin (Sigma, St. Louis, MO, USA), and plaques were manually counted after staining them with 0.2% crystal violet (Sigma, St. Louis, MO, USA).

### 2.4. Viral Passages

Vero cells were seeded in 24-well plates, aiming to reach approximately 80% the next day. For passage 1, the viral stock was diluted in PBS to obtain a multiplicity of infection (MOI) of 5. After removing the cell culture medium, cells were incubated with the viral solution at 37 °C for 1 h. Following virus adsorption, the inoculum was removed and replaced with 600 μL of the appropriate cell culture medium containing 2% FCS. At 48–72 h post infection, the supernatant was harvested and clarified by centrifugation (12,000× *g*, 5 min). The following passages were performed blindly, using 300 μL of the clarified supernatant from previous passage to infect naïve cells, followed by the same procedure. A total of 8 passages were performed. Each passage was titered by plaque assay. Six replicates were performed per virus.

### 2.5. Growth Curves

Vero cells were seeded in 24-well plates 24 h prior to infection, in order to reach 80% of confluence the next day. Virus stock was diluted in PBS to reach the intended MOI (0.01 or 1) and incubated on the cells. After one hour, the virus was removed, and cells were washed three times with PBS. Fresh medium supplemented with 2 FBS was added. At each time point, 50 μL of medium was harvested (kept at −80°) and replaced by 50 μL of fresh medium. All samples were titered together by plaque assay. Both growth curves were realized in triplicate.

### 2.6. Deep-Sequencing

RNA of 100 μL of each sample was extracted using a ZR-96 Viral RNA kit (Zymo, Research, Irvine, CA, USA) following the manufacturer’s protocol, and then eluted in 20 μL nuclease-free water. The RNA library was produced with the NEBNext Ultra II RNA Library kit (Illumina, San Diego, CA, USA), using Multiplex oligos (Illumina, San Diego, CA, USA). Libraries were quantified using the Quant-iT DNA assay kit (Thermo Fisher Scientific, Life Technologies Corporation, Grand Island, NY, USA) and diluted to 1 nM prior to sequencing on a NextSeq sequencer (Illumina, San Diego, CA, USA) with a NextSeq 500 Mid Output kit v2 (Illumina, San Diego, CA, USA) (151 cycles).

### 2.7. SHAPE-MaP of WT and D2S CHIKV RNA

For determination of CHIKV secondary structure in Vero cells, the viral stock of the WT or D2S mutant was passaged once in fresh Vero cells seeded in a T150 flask. Supernatants were harvested and viral RNA were extracted from sucrose-purified virions and analyzed as described in a previous work [35].

### 2.8. Data Analysis

#### 2.8.1. Alignment and Identification of DVGs

The BBTools suite was used to analyze the sequencing output (Bushnell B.—sourceforge.net/projects/bbmap/). BBDuk allowed us to trim for low-quality bases and adaptors, using fastq files generated from sample sequencing. Then, these were aligned to the CHIKV reference sequence (Carib—GenBank accession no. LN898104.1, IOL—GenBank accession no. AM258994) using a BBMap. To visualize the data, heatmaps showing the deletion score of each nucleotide position were generated using R. Specifically, scores were computed as the sum of the number of reads per million reads (RPM) supporting the deletion of a specific nucleotide position. For plotting start/stop breakpoints on R, deletions with lengths below 10 nucleotides were discarded.

#### 2.8.2. Correlation between DVG Nucleotide Allele Frequency and SHAPE Reactivity

From the list of deletions generated from deep sequencing data as described above, we selected all deletions of 10 or more nucleotides. Breakpoints in the 3′UTR (i.e., after nucleotide 11302) were excluded, since SHAPE MaP is less accurate in that region because of sequence repeats [38]. From this list, we broke down the data by start and stop positions and associated allele frequency (in reads per million). Positions that were both start and stop positions were pooled separately and their associated allele frequency values in reads per million were summed up. Next, we matched the raw SHAPE reactivity value to each position, and then calculated a Pearson correlation coefficient.

#### 2.8.3. Comparison of the SHAPE Reactivity of Nucleotides Used or Not Used as DVG Breakpoints

Start and stop positions of DVGs bearing deletions of at least 10 nucleotides and generated in Vero cells were matched with SHAPE reactivity values. Nucleotides that were used as breakpoints or left unused were plotted according to their SHAPE reactivity values. A *t*-test, with Sidak’s multiple comparisons test, was used to compare means.

#### 2.8.4. Comparison of DGVs Generated in WT and D2S Mutant

From the list of deletions generated as described in the previous section, we selected the DVGs lacking at least 20 nucleotides. We pooled all replicates and passages together for a given virus (WT or D2S mutant; see below). We then separated all DVGs into two groups, depending on whether their start breakpoints were located before or after position 5000. Then, we compared the number of samples containing a given deletion (exactly the same start and stop positions) in both WT and mutant, either on a 2D plot or by displaying the distribution of their absolute differences.

#### 2.8.5. DVG Entropy

From the DVG list generated as described above, we calculated an equivalent of Shannon entropy for deletions for each sample as—Σ *p_i_* log_2(_*p_i_*), where *pi* is the ratio of the number of reads supporting the junction *i* over the total number of reads supporting all junctions.

#### 2.8.6. Statistical Analysis

Statistical analyses were performed using R version 4.2.1 (CRAN) or Prism version 8 (Graphpad). All tests were two-sided, and a *p* value < 0.05 was considered significant. Correction for multiple testing was performed using Bonferroni’s method or Sidak’s method.

## 3. Results

### 3.1. Preferential Generation of DVG Breakpoints at Unpaired Nucleotides

Because DVGs highjack the machinery of full-length virus for replication and/or transmission, they preferentially accumulate at high multiplicity of infection (MOI) when each cell is co-infected with multiple viral particles [39,40]. To enrich for DVGs, we performed serial passages at high MOI in triplicates in Vero cells (Appendix A). At each passage, half of the supernatant was used to infect cells for the next passage and half was used for next-generation sequencing and bioinformatic analysis using an in-house pipeline [9,10,18]. DVGs were defined by their start and stop breakpoints, corresponding to the first and the last nucleotide deleted [9,10,18] (Figure 1A, hereinafter conjointly termed “breakpoint nucleotides”). As described in a previous work [10], classifying DVGs using the start and the stop positions of the truncation identifies clusters of DVGs on the genome (see clusters A, B and C in Figure 1A,B).

We observed no obvious sequence homologies between the start and stop regions for the commonly deleted regions which would suggest template-switching to a near-homologous region. Additionally, there are currently no reported RNA binding proteins associated with these regions that would bring the start and stop regions into close proximity during replication, allowing the RdRp to skip over the deleted nucleotides. Therefore, we hypothesized that RNA secondary structure in breakpoint regions may play a role in generating these specific DVGs.

We used data generated from our prior SHAPE-MaP analysis of the CHIKV genomic RNA [35] to determine the RNA secondary structure of the viral genome around the breakpoints. Reactive nucleotides, or nucleotides that are more flexible, are colored in orange, while unreactive nucleotides, those that are protected from chemical modification due to base pairing, are colored black, in the example region of the cluster A and B start breakpoints (nucleotide 300 to 819, Figure 1C). To determine if there was a relationship between a specific nucleotide used as a DVG break site in CHIKV and the flexibility of that nucleotide, we assessed whether the SHAPE reactivity of the breakpoint nucleotides differed from the SHAPE reactivity of unused nucleotides after 1 passage in Vero cells (Figure 1D). A median SHAPE reactivity over a 5-nucleotide window was used to capture the flexibility of a hyper-local region around the breakpoint. The average median SHAPE reactivity of nucleotides used as breakpoints remains higher than those not used as breakpoints over all passages, assuming each breakpoint was once originally generated from a full-length genome (Appendix A).

Next, we analyzed the correlation between the frequency at which a nucleotide was used as a breakpoint (number of reads with this nucleotide as a breakpoint over the total number of reads) with the nucleotide’s median SHAPE reactivity (i.e., its probability of being paired when low or unpaired when high, Figure 1C), using a 5-nucleotide window. We used pooled data generated from all replicates over 8 passages (Figure 1E) and focused our analysis on DVGs presenting a deletion of at least 10 nucleotides in order to exclude all small deletions that might be generated by a short slippage of the viral polymerase rather than a canonical event of non-homologous recombination of RNA molecules. We uncovered an enrichment of unpaired nucleotides in those used as DVG breakpoints, even though the relationship is weak (Figure 1E, Pearson correlation *r* = 0.066, *p* = 2.7. 10–11).

Together, these results suggest that CHIKV DVG breakpoints preferentially occur at more flexible nucleotides, which are more likely to be unpaired, although flexibility alone is not sufficient to predict a breaksite.

### 3.2. CHIKV-D2S Mutant with Disrupted Secondary Structure Is Viable in Cell Culture

To experimentally test the impact of secondary structure on the generation of DVGs, we generated a mutant with a disrupted secondary structure (D2S) around a subset of important DVG breakpoints found in CHIKV passages in Vero cells (Figure 1A and Figure 2). We designed the D2S mutant so that its secondary structure would be disrupted around the DVG breakpoints of clusters A, B and C (all located in the first half of the genome, before nucleotide 5000) without impacting the secondary structure located after nucleotide 5000 (Figure 2A). Hence, we modified 3 regions with a total of 76 synonymous mutations generated using the program CodonShuffle with the dn231 algorithm (Jorge, Mills, et Lauring 2015): 19 mutations were introduced in the first region, covering nucleotides 380 to 730 (where cluster A and B start breakpoints are located); 35 mutations were between nucleotides 2810 and 3190 (comprising cluster A stop breakpoints and cluster C start breakpoints); and 22 mutations were between nucleotides 3800 and 3950 (comprising cluster C stop breakpoints) (Figure 2A,B; the list of mutations is rendered in Appendix A). The mutations were selected with the aim of modifying the predicted RNA structure as much as possible without changing the amino acid sequence and codon usage (Figure 2B). To assess whether D2S showed decreased fitness (i.e., slower replication kinetics) in these settings compared to WT virus, we performed one-step and multi-step growth curve analyses in Vero cells. The D2S mutant displayed slower growth kinetics with a ten-fold disadvantage compared to WT (*p* < 0.0001; Figure 2C). However, in the one-step growth curve, at 48 h post-infection, D2S and WT had similar titers in Vero cells. Of note, both Sanger sequencing and deep sequencing results confirmed that none of the mutations introduced in D2S reverted throughout the series of passages.

### 3.3. Altered DVG Generation in CHIKV D2S, Compared to WT Virus

To interrogate the role of genomic RNA secondary structures in the generation of DVGs, we compared the generation of DVGs between the WT CHIKV and D2S mutant after high-MOI passaging (Appendix A). DVG sequences were characterized by next-generation sequencing and visualized depending on their start and stop positions. WT and D2S tended to have close global distributions of deletions (Figure 3A). This finding was confirmed by mapping the probability of each nucleotide position to be deleted (Appendix A). To explore the diversity of DVGs generated by wild-type and mutant virus, we calculated the equivalent of a Shannon entropy for DVG breakpoints. For both the WT and D2S mutant, entropy decreased through passages, suggesting a decrease in DVG diversity over time. Entropy was significantly higher in the D2S mutant compared to the WT throughout the passages (*p* < 0.001, Figure 3B), illustrating that the D2S mutant generated a more diverse pool of DVGs, compared to WT virus.

### 3.4. DVG Generation Is Governed by RNA Secondary Structures

To formally evaluate the importance of secondary structure in DVG generation, we separated DVGs depending on their sequence. A first group was made with DVGs showing a start breakpoint before nucleotide 5000, and a second group with start breakpoints after nucleotide 5000. The region before 5000 contains the three regions where we disrupted secondary structures, as well as the start breakpoints of the DVG clusters A, B and C, which are predominant in Vero cells (Figure 1A and Appendix A). On the contrary, the region after nucleotide 5000 should display RNA secondary structures equal or similar to WT, since no mutations have been introduced there (Figure 1A). We listed all the different deletions present in the WT and D2S samples and plotted the number of samples that displayed each individual deletion in WT (x axis) or D2S (y axis). DVGs that appeared predominantly, or exclusively, in one virus would stack along the x and y axes, while DVGs that were common between WT and D2S samples would appear closer to or along the diagonal. We found that DVGs before nucleotide 5000 were often found only in one virus and not the other (Figure 4A), while DVG landscapes were more similar between the two viruses after nucleotide 5000.

To precisely assess this difference in the DVG landscape between the two viruses in these regions, we computed, for each specific deletion, the absolute difference between the number of samples supporting these breakpoints in the WT and the DS2 viruses. Then, we computed the number of specific DVGs supporting a specific absolute-count difference (Figure 4B). The distributions of these differences significantly differed between DVGs before and after nucleotide 5000, with fewer similar DVGs (between WT and DS2 mutants) in the first part than in the second part of the genome (*p* < 0.0001, Kolmogorov–Smirnov test and unpaired *t*-test).

We then performed SHAPE-MaP analysis on the D2S mutant to confirm that our mutations disrupted the secondary structure of the virus, as predicted. We calculated the change in SHAPE at each nucleotide between WT and D2S. If the mutations disrupted local secondary structure but maintained the RNA secondary structure of distal genome regions, there should be a larger change in SHAPE between WT and D2S in and near mutated regions than observed in and near distal regions. Indeed, the median change in SHAPE reactivity for nucleotides outside the modified regions was significantly smaller than for nucleotides within the modified region, corroborating the interest of the D2S mutant in this experiment (Figure 4C). Then, we looked at SHAPE reactivity values of nucleotides used as breakpoints, considering the frequency of their use, and observed a significant correlation (*r* = 0.053, *p* = 6.9 × 10^−10^, Figure 4D), as already seen with WT (Figure 1E). Interestingly, this correlation was not significant when using WT SHAPE reactivity values as a reference for D2S DVGs, reinforcing our observations. In line with what was observed with WT, the nucleotides used as breakpoints during D2S infection after a single passage had higher median SHAPE reactivity than nucleotides that were never used as breakpoints (Figure 1E). Together, these data suggest that in both D2S mutant and WT virus, secondary structures influence DVG generation through non-homologous recombination.

## 4. Discussion

Advances in next-generation sequencing technologies allow a more thorough exploration of viral RNA secondary structures; it is already known that RNA secondary structures are key for genome replication and translation in alphaviruses [23,35,41,42,43], and in splicing and viral gene expression in HIV [44]. RNA secondary structures have also been described as being mediators of homologous recombination [24,26,45,46]. Concomitantly, DVGs have gathered greater interest over the last decade for their potential use as antiviral molecules [11,12], yet the rules governing their generation are still unclear. Our work supports the notion that RNA secondary structures influence how CHIKV DVGs are generated.

To examine this association, we used RNA secondary structures modeled using SHAPE-MaP analyses of the CHIKV genome, a source which is, to date, the most precise prediction of RNA structures currently attainable [35]. One of the specific strengths of our analysis lies in using the median SHAPE reactivity of a 5-nucleotide region around the nucleotide of interest, while studies of SHAPE reactivity generally rely on using the median SHAPE reactivity of a 50-nucleotide window [23,47,48]. Indeed, although DVG recombination sites can occur in highly structured regions, recombination could also preferentially happen on the unpaired, rather than paired, nucleotides of such secondary structures. We chose to exclude the 3′UTR region from analysis, even though CHIKV DVG breakpoints often occur in this region [10]; many sequence repeats complicate the analysis and decrease prediction performance compared to genomic RNA [38]. Finally, our initial analyses pooled DVGs from all passages combined, which makes it impossible to differentiate newly generated DVGs from DVGs that are replicated from a previous DVG template.

The reported correlation between DVG breakpoint probability and SHAPE reactivity, although significant, remains moderate. Our data therefore suggest that if RNA structure is an important driver of DVG formation, it remains complex and coupled to other determinants. This finding is consistent with what was observed in our previous work on CHIKV DVGs [10]: DVGs can be produced in various species, but their sequence and abundance depend on both the cellular environment and the viral genome. Other determinants may include RNA–cellular or RNA–viral protein interactions, long-distance RNA interactions (tertiary structures or interaction with cellular RNA), sequence homology around breakpoints, cellular ionic concentrations, innate immunity pressure, or location of viral replication.

We were able to experimentally test the putative role of secondary structure in DVG formation by creating a mutant with altered secondary structure. To our knowledge, only two studies have disrupted secondary structures of alphaviruses at the nucleotide level without modifying the amino-acid sequence using CodonShuffle [37]. Using SINV and CHIKV, the authors introduced mutations in the RNA secondary structures and confirmed that these mutants had growth defects, a finding arguing for the importance of secondary structures in these regions [23,35]. One pitfall of this method is the absence of controls bearing the same number of mutations without disrupting the secondary structure, and that would behave as WT. In our case, because any slight nucleotide change could cause recombination changes in the hypothetical DVG breakpoints being shaped by the RNA structures, it is close to impossible to design such a control. Nevertheless, this drawback was compensated for by the fact that we split the genome in two: a first region, before the 5000th nucleotide, where the RNA secondary structure was greatly disrupted; and a second region, after nucleotide 5000, where secondary structure was similar to WT, providing some internal control. Notwithstanding this, because disrupting secondary structures could also impact long-distance interactions (tertiary structures) or protein interactions, recombination areas could still be affected in the second half of the genome, which may explain the remaining differences between DS2 and WT DVG landscapes in this unchanged region.

Even though DVGs with large deletions generated from WT and D2S mutants do not present exactly the same breakpoints, both viruses tend to delete the same regions when passaged in the same cell type. However, the D2S mutant seems to fumble when generating DVGs: it creates more diverse DVGs, and at lower frequencies. Environmental pressure may still drive a particular kind of DVG, while the abnormal secondary structure makes it harder to select for specific DVGs. This may imply that the WT genome has evolved to maintain, or avoid the generation of, certain DVGs.

In conclusion, although other determinants are involved, RNA secondary structures do drive CHIKV DVG formation. Further work is needed to narrow down which specific RNA structures are involved in DVG generation, and to pinpoint other determinants such as sequence homology around breakpoints, RNA protein interaction, or RNA tertiary structures. A broader knowledge of these mechanisms could help develop prediction tools for DVG generation in different viruses and enable researchers to better genetically engineer genomes and DVGs for therapeutic purposes.

## Figures and Tables

**Figure 1 microorganisms-12-01794-f001:**
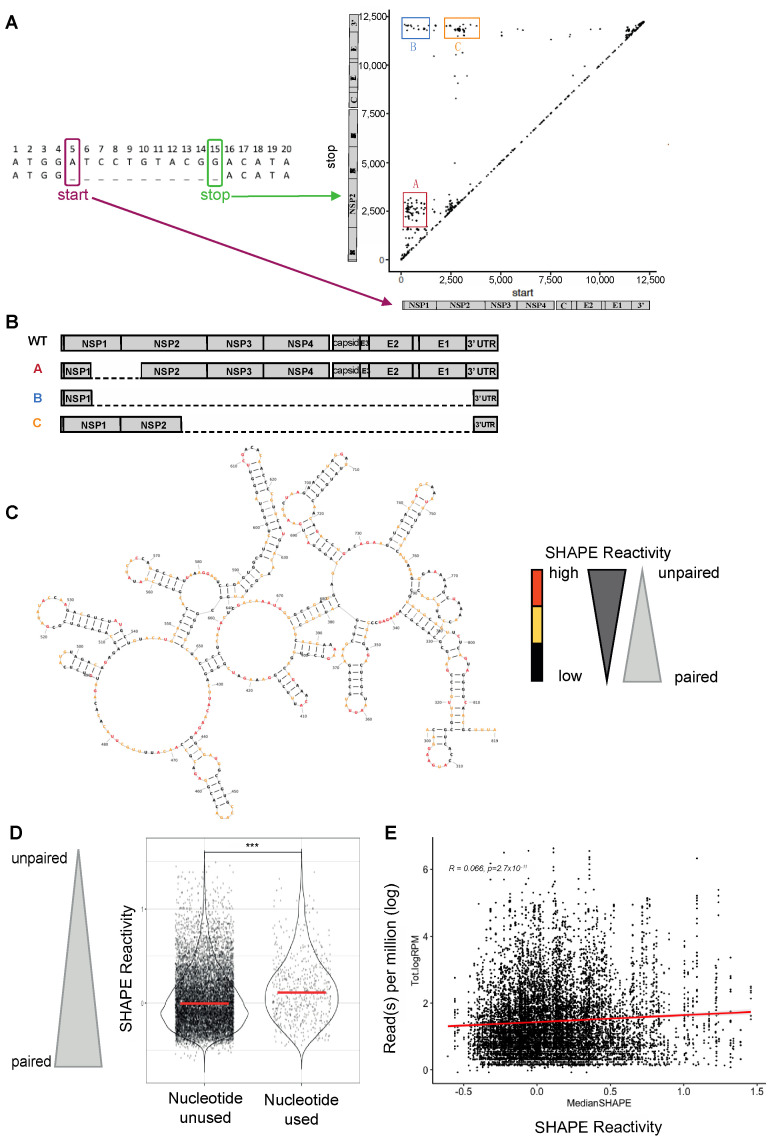
CHIKV genome secondary structures correlated with DVG generation. (**A**) Schematic of start and stop breakpoints of DVGs generated by high-MOI passages of CHIKV in Vero cells. Start (x axis) and stop (y axis) positions of the breakpoints of DVGs are plotted. The different clusters are called A, B, and C. (**B**) Schematic of WT CHIKV genome. (**C**) CHIKV Carib secondary structures from the SHAPE-MaP analysis [35] around the start breakpoints of DVGs A and B (Nucleotides 300 to 819). Nucleotides are depicted in black (low), red (high) or yellow (intermediate), depending on their SHAPE reactivity. (**D**) In first passage, nucleotides that were used as breakpoints (start and/or stop) or left unused were plotted according to their median SHAPE reactivity value around a 5-nucleotide interval. *** *p* < 0.001 (unpaired *t*-test). (**E**) Reads-per-million values for each nucleotide position used as a start and/or stop breakpoint (start or stop), matched with their respective rolling median SHAPE reactivity (of a 5-nucleotide window) over all passages. Values *r* and *p* denote Pearson’s correlation coefficient and its associated *p*-value.

**Figure 2 microorganisms-12-01794-f002:**
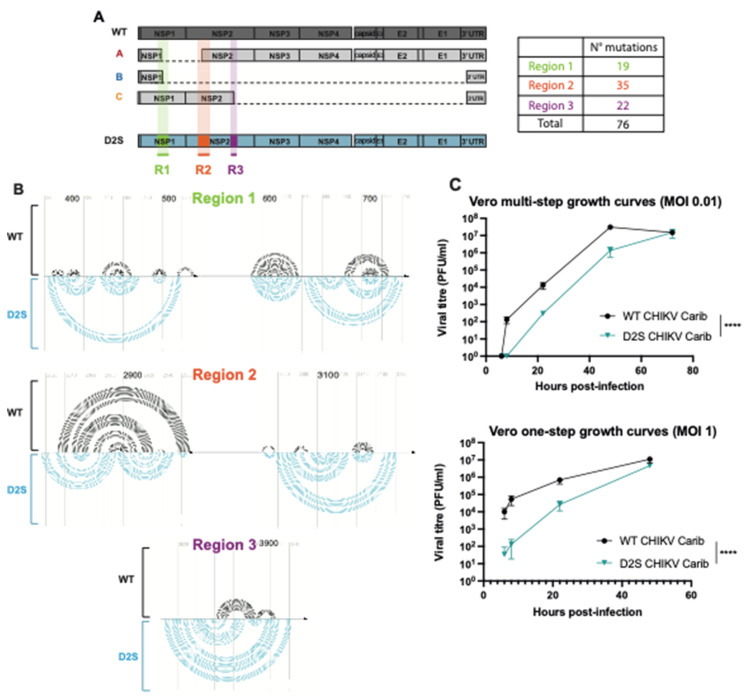
Engineering a mutant with disrupted secondary structure around the breakpoints of clusters A, B and C. (**A**) Schematic of WT CHIKV, and the DVGs from clusters A, B and C and the D2S mutant, highlighting the 3 mutated regions and the number of silent mutations inserted. (**B**) Predicted secondary structures, determined using SHAPE-MaP data for WT virus (black lines) or mFold software (UNAfold, http://www.unafold.org) for D2S (blue lines), after introducing mutations around regions 1, 2 and 3. The black arrows represent CHIKV genome parts, and each line represents an interaction between 2 nucleotides. Several lines forming an arc form a hairpin. (**C**) Growth curves of D2S CHIKV (blue line) and WT CHIKV (black line) at MOI 0.01 (multi-step growth curve) and 1 (one-step growth curve) in Vero cells. Bars represent mean ± SD, *n* = 3 biological replicates; **** *p* < 0.0001 (two-way ANOVA with Bonferroni post-test).

**Figure 3 microorganisms-12-01794-f003:**
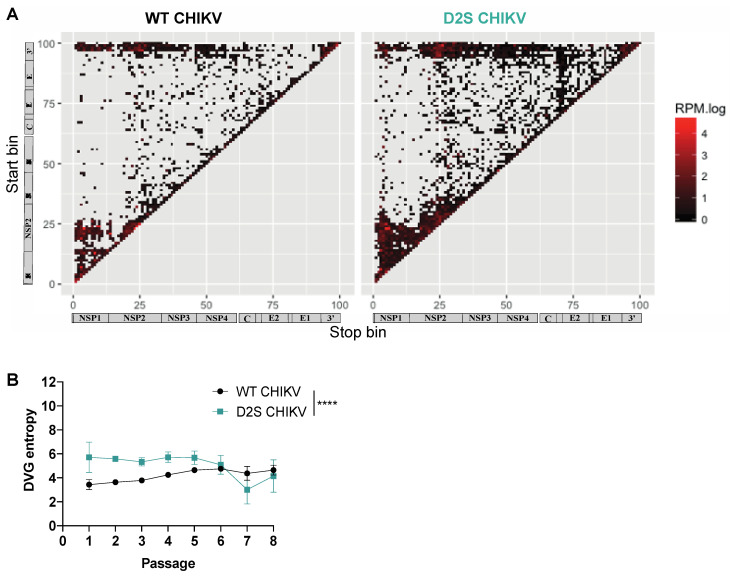
D2S mutant generates a greater diversity of DVGs. (**A**) DVGs were mapped on their start and stop positions, with colors representing summed frequencies of the DVGs with the same start and stop bins. (**B**) DVG entropy of WT and D2S mutant over serial passaging. **** *p* < 0.001 (unpaired *t*-test).

**Figure 4 microorganisms-12-01794-f004:**
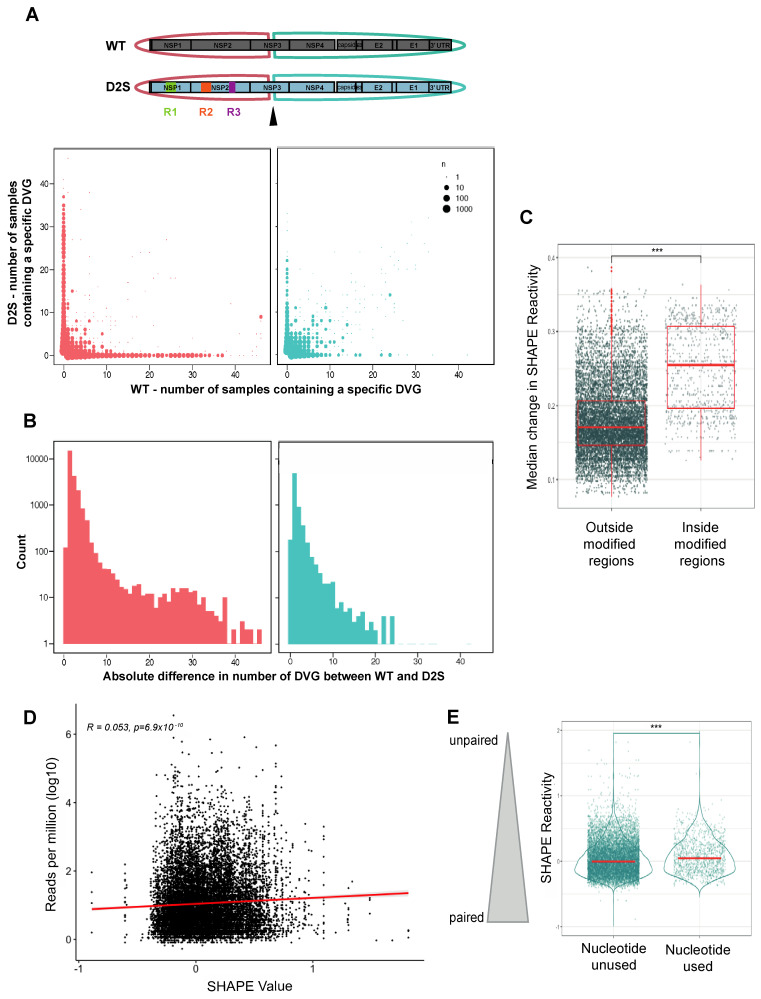
DVG generation is governed by secondary structures. (**A**) All DVG (start, stop) positions for pairs found in WT and D2S are listed and associated with the number of DVGs supporting this pair for each virus over all replicates. We separated DVGs that had a start position before (red) or after (blue) the 5000th nucleotide. (**B**) Distributions of the absolute difference between DVG counts in WT vs. DS2 viruses for each (start, stop) pair. *p* < 0.0001 (Kolmogorov–Smirnov test and unpaired *t*-test). (**C**) Median change in SHAPE reactivity of the D2S mutant compared to WT, inside or outside modified regions (R1, R2 and R3 pooled together), *** *p* < 0.001 (unpaired *t*-test). (**D**) Reads-per-million values for each nucleotide position used as a start and/or stop breakpoint were matched with the SHAPE reactivity value for the nucleotide. Values *r* and *p* denote Pearson’s correlation coefficient and its associated *p*-value. (**E**), Nucleotides that were used as breakpoints (start and/or stop) or left unused by D2S mutant over all passages were plotted according to their SHAPE reactivity value (determined in D2S mutant). *** *p* < 0.001 (unpaired *t*-test).

## Data Availability

Part of the datasets presented in this article are not readily available due to technical limitations. Requests to access the datasets should be directed to the corresponding author, L.I.L.

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
