# Peer review of "Chikungunya Virus RNA Secondary Structures Impact Defective Viral Genome Production"

_microorganisms, 2024, doi:10.3390/microorganisms12091794_

Round 1
Reviewer 1 Report
Comments and Suggestions for Authors
This is a very clever and worthwhile exploration of the impact of DVGs on the viability of the CHIKV virus. This has wider applicability than simply to Alphaviruses. You mention one source related to HIV. In our HIV work, we also are interested in exploring the potential of DVGs in creating defective, non-replicable viruses.
You have some issues with English usage, where I could not understand clearly what you were trying to convey. An additional thorough language edit is the only recommendation I would make.
Very impressive.
Comments on the Quality of English LanguageSee above .
Author Response
Comment 1 :
This is a very clever and worthwhile exploration of the impact of DVGs on the viability of the CHIKV virus. This has wider applicability than simply to Alphaviruses. You mention one source related to HIV. In our HIV work, we also are interested in exploring the potential of DVGs in creating defective, non-replicable viruses.
You have some issues with English usage, where I could not understand clearly what you were trying to convey. An additional thorough language edit is the only recommendation I would make.
Very impressive.
Response 1 : Thank you for your feedback. This manuscript has been corrected by native english speakers. Few modifications have been done.
Reviewer 2 Report
Comments and Suggestions for Authors
This paper addresses an important and interesting problem—the influence of viral RNA secondary structures on the production of CHIKV DVGs. The authors experimentally map RNA secondary structures of CHIKV genome using selective 2′-hydroxyl acylation analyzed by primer extension and mutational profiling (SHAPE-MaP). Ultimately, they showed that, in infected mammalian cells, the higher the probability for a given nucleotide to be unpaired, the higher the probability for it to be a DVG break-point. Importantly, the authors correlate the nature of the DVGs produced by the D2S mutant to its RNA genomic structure determined by SHAPE-MaP, and directly demonstrated that RNA secondary structures are involved in the regulation of non-homologous recombination and DVG generation. Overall, the article is well organized and well- written. However, some minor issues still needed to be improved:
(1) There are some writing errors that need to be improved, such as line17 “in-tensively”, 22 “struc-ture”, 23“proba-bility”, 24 “second-ary”, 27 “corre-lating”.
(2) Line 68 “nSP1” should be “nsp1/NSP1.”
(3) DVGs have been documented to influence viral replication and the activation of the immune system. In this study, compared to wild-type virus, CHIKV-D2S presents altered DVG generation and significantly reduces its replication ability. Is there a correlation between the changes generated by DVG and the decrease in replication ability for CHIKV-D2S?
(4) DVGs preferentially accumulate at high multiplicity of infection (MOI). To enrich for DVGs, the authors performed serial passages at high MOI in triplicates in Vero cells. At each passage, half of the supernatant was used to infect cells for the next passage and half was used for next generation sequencing and bioinformatic analysis using an in-house pipeline. ??? In this experiment, did the author accurately measure the initial viral inoculation titers during each passage to ensure that WT- and CHIKV D2S-group were the same.
Comments on the Quality of English LanguageThe overall writing is good, but some minor writing errors in the article need to be corrected.
Author Response
Comment 1: There are some writing errors that need to be improved, such as line17 “in-tensively”, 22 “struc-ture”, 23“proba-bility”, 24 “second-ary”, 27 “corre-lating”.
Response 1: This modification has been done.
Comment 2: Line 68 “nSP1” should be “nsp1/NSP1.”
Response 2 :This modification has been done.
Comment 3: DVGs have been documented to influence viral replication and the activation of the immune system. In this study, compared to wild-type virus, CHIKV-D2S presents altered DVG generation and significantly reduces its replication ability. Is there a correlation between the changes generated by DVG and the decrease in replication ability for CHIKV-D2S?
Response 3: Although we can observe a correlation between the changes in DVG production and a slower kinetic we can not conclude to a causal link. Previous work in our team and others has shown that highest abundancy in DVG impact viral replication (levi et al 2021, Rezelj et al 2021, Rezelj et al 2018). Here, it is the diversity of DVG produced that is mainly impacted. The disruption of the secondary structure could be by itself responsible of this slower kinetic, independently of DVG production.
Comment 4: DVGs preferentially accumulate at high multiplicity of infection (MOI). To enrich for DVGs, the authors performed serial passages at high MOI in triplicates in Vero cells. At each passage, half of the supernatant was used to infect cells for the next passage and half was used for next generation sequencing and bioinformatic analysis using an in-house pipeline. ??? In this experiment, did the author accurately measure the initial viral inoculation titers during each passage to ensure that WT- and CHIKV D2S-group were the same.
Response 4: As mentioned in methods, we realized blind passages, the titration was done after the passage occured. We decided to do so since we wanted the highest MOI possible to generate the maximum DVGs. Retrospectively, we saw that MOI were equivalent between WT and D2S mutant (suppl fig 1A). This is probably explained by the fact that also the mutant show slower kinetics, it has the same titer at H48 (time used to proceed to next passage, fig 2C).
Reviewer 3 Report
Comments and Suggestions for Authors
In the manuscript by Levi et al., the authors aimed to investigate how viral RNA secondary structures affect the production of Chikungunya virus (CHIKV) defective viral genomes (DVGs). Using SHAPE-MaP, which combines chemical labeling with next-generation sequencing, researchers mapped the RNA secondary structures of the CHIKV genome. They found that DVG generation is more likely at unpaired nucleotides within these structures. A CHIKV mutant (CHIKV-D2S) with synonymous changes disrupting RNA secondary structure showed altered DVG production, linking changes in RNA secondary structure to DVG generation. This work demonstrates the influence of RNA secondary structure on CHIKV DVG production during replication.
Overall, the study is interesting and may have implications for engineering viruses like CHIKV and others for therapeutic purposes. The findings are clearly presented and data supports the conclusions. I have only a few comments which are given below.
Minor comments
1. It would be interesting to see whether other CHIKV strains under different tissue tropism in vitro and in vivo exhibit similar findings, which may guide the development of broad-spectrum therapeutics.
2. This study lacks the functional relevance of the DVGs. To what extent these DVGs are functionally relevant to CHIKV-associated infections and disease outcomes?
3. DVGs can also be harnessed as vaccine adjuvants. Can their findings be transformed into developing vaccines against CHIKV?
Author Response
Comment 1: It would be interesting to see whether other CHIKV strains under different tissue tropism in vitro and in vivo exhibit similar findings, which may guide the development of broad-spectrum therapeutics.
Response 1: Thank you for this interesting point. We decided to work with the Caribbean strain since the secondary structure using SHAPE MAP was already known. This work has been done on mosquito cells as well and show similar results but with a less stringent phenotype. However for clarity we decided not to include these results (unpublished data).
Comment 2: This study lacks the functional relevance of the DVGs. To what extent these DVGs are functionally relevant to CHIKV-associated infections and disease outcomes?
Response 2: This is out of the scope of the paper. This work is an experimental model in which the generation of DVGs is the highest possible in order to highlight the impact of secondary structures on DVG generation. Although this question is crucial, our design does not permit to answer it.
Comment 3: DVGs can also be harnessed as vaccine adjuvants. Can their findings be transformed into developing vaccines against CHIKV?
Response 3: A better understanding of DVG production can help generating an attenuated virus capable of generating more DVGs that would interact with wildtype virus replication. This could be useful as a vaccine but more probably as an antiviral treatment. This field still needs development in order to achieve such a goal.
Reviewer 4 Report
Comments and Suggestions for Authors
In this study, authors investigated the RNA structure of CHIKV and its influence on defective viral genomes using SHAPE-MaP model by demonstrating disrupted RNA structures. They showed that unpaired nucleotides in the secondary structure of the RNA genome contribute to the generation of defective viral genomes.
Supplemental figures and tables are missing. Please attach the missing figures and tables.
In Fig 1c, CHIKV Carib secondary structures from the SHAPE-MaP analysis have been used and cited from Madden et al., JVI, 2020. Please state that you have permission to use published data or figures/images from the journal you have previously submitted.
The quality of the images is too low; please enhance the DPI per the journal's suggestion.
Please cite Fig 3B in the text or remove panel B.
Author Response
Comment 1: Supplemental figures and tables are missing. Please attach the missing figures and tables.
Response 1: These figures have been attached to the initial submission.They will be attached to final submission.
Comment 2: In Fig 1c, CHIKV Carib secondary structures from the SHAPE-MaP analysis have been used and cited from Madden et al., JVI, 2020. Please state that you have permission to use published data or figures/images from the journal you have previously submitted.
Response 2: Although this figure has been generated from Emily Madden’s work published in JVI in 2020, the figure 1C itself is a new one never published before.
Comment 3: The quality of the images is too low; please enhance the DPI per the journal's suggestion.
Response 3: High quality files have been provided to the Editor. This will be seen with the editor.
Comment 4: Please cite Fig 3B in the text or remove panel B.
Response 4: This figure is already cited line 331/332.